# A Study on the Knowledge, Attitudes, and Behaviors of Pregnant Women Regarding HIV and Routine Rapid Testing: An Assessment in a High-Risk Marginal Area

**DOI:** 10.3390/healthcare9070793

**Published:** 2021-06-24

**Authors:** Leila Jahangiry, Zahra Aliyari, Koen Ponnet

**Affiliations:** 1Tabriz Health Services Management Research Center, Tabriz University of Medical Sciences, Tabriz 5166/15731, Iran; 2Medical Education Research Center, Health Management and Safety Promotion Research Institute, Tabriz University of Medical Sciences, Tabriz 5166/15731, Iran; zahraaliyari2016@gmail.com; 3Department of Communication Sciences, Imec-Mict-Ghent University, 9000 Gent, Belgium

**Keywords:** knowledge, attitude, behavior, pregnant women, HIV

## Abstract

Acquired immunodeficiency syndrome (AIDS) is one of the main obstacles to communities’ development. The disease mostly involves active and productive population groups. This study aimed to determine the knowledge, attitudes, and behaviors of pregnant women regarding HIV prevention and rapid HIV tests. Pregnant women who were referred to the local health centers and who were willing to participate in the study were interviewed. To collect data, a standard questionnaire was used among 200 pregnant women in eight local health centers of Kermanshah, Iran. The survey contained 50 questions on demographic characteristics and the knowledge, attitudes, and behaviors regarding HIV/AIDS prevention and rapid tests for pregnant women. Although the majority (82.5%) of the pregnant women knew that mother-to-child HIV transmission during pregnancy was possible, fewer than half (48.2%) of them knew that HIV can be transmitted from mother to child through breastfeeding. Only 22.5% of pregnant women knew that a Cesarean section for HIV-positive mothers is recommended. The mean attitudes of pregnant women toward HIV prevention and HIV rapid testing were 4.5 (SD = 0.4) and 4 (SD = 0.3), respectively. Of the women, 11.5% had participated in an HIV rapid test counseling class, and 25.5% had participated in HIV education and counseling classes. The low knowledge of mothers regarding HIV transmission highlights the need for education and counseling classes and campaigns to improve knowledge and behaviors related to HIV prevention, especially during pregnancy for women in marginal regions.

## 1. Introduction

Acquired immunodeficiency syndrome (AIDS) is a lethal infectious disease and the fourth leading cause of mortality worldwide. It is one of the main obstacles to communities’ development and mostly involves active and productive populations [1]. According to a World Health Organization (WHO) report, more than 70 million people have been infected and more than 35 million have died of this infection since the onset of the HIV infection epidemic [2]. At the end of 2016, 36.7 million people worldwide were living with AIDS; of these, about 41% were women at a reproductive age between 15 and 49 years old [3]. At the same time, approximately 3.4 million children under the age of 15 years old were infected. At the end of 2016, national reports from Iran identified 34,846 people infected with HIV. Currently, it has been estimated that over 76,000 people in Iran live with HIV [4], and new HIV infections increased by 21% from 2010 to 2016 [5,6].

The most important source of HIV infection among children and newborns is mother-to-child transmission during pregnancy, infection during childbirth, and breastfeeding [7]. Evidence shows that the risk of transmission varies at different stages: from 5% to 10% during pregnancy, 10% to 20% during delivery, and 10% to 20% during infant feeding [8]. There is also evidence that increasing incidences of HIV in pregnant women will lead to increased incidences of HIV in children [9]. According to some studies, the risk of contracting HIV is the highest among children during the first month after birth and the lactating period [10]. Therefore, the Iranian Ministry of Health is trying to prevent AIDS by focusing on blood safety, planning harm reduction programs for people who inject drugs, and setting up programs to end mother-to-child transmission of HIV by means of routinely performing HIV rapid testing (RT) [6].

Despite substantial improvements in HIV surveillance in Iran [11], people with low economic status that live in marginal areas of metropolitan cities often experience major difficulties to accessing HIV healthcare [12]. Furthermore, these high-risk groups are often not recognized by the government. They receive limited education on HIV prevention, and no education campaigns exist for them in Iran [13]. There is also strong evidence that people in marginalized areas, often migrant people with low socioeconomic status, are less likely to enter HIV prevention programs, making the success rate of HIV prevention programs complicated [14,15]. 

There are limited studies regarding HIV-related knowledge, RT behaviors, and beliefs among high-risk population groups. A study from the United States among internal medicine found that fewer than one-third of physicians always or usually performed routine testing [16]. According to the WHO, the acceptability of HIV testing is increasing, but still about half of the people located in poor regions and living with HIV are unaware of their status, which is particularly the case among pregnant women [17]. As such, there is an urgent need for effective programs to control HIV in marginal areas. One of the key goals of the Iranian HIV prevention programs in which RT is conducted includes increasing the HIV testing rates among pregnant women. Indeed, several authors have claimed that pregnant women should be involved in HIV prevention programs and have to be aware of the importance of HIV counseling and testing, not only for themselves but also for their infants due to the probability of mother-to-child transmission during pregnancy, childbirth, and breastfeeding [18,19]. 

The purpose of this study was to assess the knowledge, attitudes, and behaviors of pregnant Iranian women regarding HIV prevention and rapid testing. Women in marginal regions have often been neglected; therefore, this study focused on pregnant women who visited the health centers of Kermanshah, Iran.

## 2. Methods

### 2.1. Study Area 

The study participants were pregnant women in the marginal area of Kermanshah, Iran, populated by approximately 851,000 people. Currently, Kermanshah is the fourth largest city of Iran in terms of the marginalized migrant population. In this area, the low socioeconomic context of the society influences the population’s health conditions [18]. 

### 2.2. Study Design and Sample Size

In total, 208 pregnant women in eight local health centers were randomly enrolled in this cross-sectional study during the period of 23 September 2017–20 February 2018. A cross-sectional design was applied, given that we wanted to assess the knowledge and attitudes as disease traits. Eight women were removed from the analyses because they did not complete the full interview. Participants were randomly selected pregnant women who were referred to the local health centers and who were willing to participate in the study. Eligibility criteria were (a) being pregnant; (b) living in the marginal area of Kermanshah; and (c) receiving pregnancy care. Written informed consent was obtained before the participants were enrolled in the study. All interviews were conducted face-to-face at the healthcare centers by the second author of this article (Z.A.). The sample size for the study was calculated using the following formula (*n* = Z^2^(1 − α/2) P (1 − P)/d2), where z(1 − α/2) for 95% confidence intervals was 3.84, P = 0.6 [18], and d = 0.72. As such, based on the estimated sample size, at least 185 pregnant women were required for this study to have a power of 80% at the 5% significant level. However, in practice, 200 pregnant women were enrolled in the study.

### 2.3. Study Procedure

A standardized questionnaire was developed by the research team specifically for the purpose of this study. The questionnaire contained 50 questions on demographic characteristics and knowledge, attitudes, and behaviors relating to HIV/AIDS prevention and rapid tests for pregnant women. Questions were drawn from existing tools [20,21,22,23,24] and divided into three dimensions, including knowledge, attitudes, and behaviors regarding HIV prevention and HIV RT on the bases of essential specific issues for pregnant women. 

The dimensions of the questionnaire were qualitatively assessed based on a conceptual framework designed by a panel of five health professionals. The health professional panel determined whether the contents of the questionnaire were relevant to the conceptual framework. The attitude dimension was evaluated based on the validity and reliability indexes before data collection, and was found to be a valid and reliable measure. The Cronbach’s alpha coefficients for the separate dimensions ranged from 0.65 to 0.89. Additionally, the reliability of the dimensions was assessed by intraclass correlation coefficients (ICC) and showed satisfactory results (ICC ranged from 0.75 to 0.82) among (*n* = 10) pregnant women. The questionnaire was pilot-tested with a sample of 20 respondents.

Knowledge. Questions on knowledge included 14 self-developed items. Each item had to be scored on a three-point scale (“Yes,” “No,” “I do not know”). The items were related to patients’ knowledge about the causes of HIV/AIDS, modes of transmission, mother-to-child transmission, and preventive behavior during pregnancy and breastfeeding, as well as on HIV RT. A sample item is: “Is breastfeeding recommended for HIV-positive mothers?” An aggregated score was created by summing up all correct answers, resulting in a score from 0 to 14, with a higher score indicating greater knowledge.

Attitudes. The attitudes were measured through 26 self-constructed items on a five-point Likert scale (ranging from 1 = strongly agree to 5 = strongly disagree). The attitudes scale measured pregnant women’s beliefs about HIV/AIDS causes, transmission, mother-to-child prevention, routine HIV RT, and HIV counseling (score range: 1–5). A higher score indicated that people had more positive attitudes toward HIV prevention (15 items) and HIV RT (11 items). Sample items for attitudes toward HIV prevention and RT are: “I think taking medicines like anti-viral drugs during pregnancy decreases the risk of mother-to-child HIV transmission” and “I think all pregnant women have to undergo an HIV rapid test during pregnancy.” 

Behavior. Behavior was assessed using 9 binary items (yes/no). A sample item is: “Have you ever participated in HIV RT counseling?” An aggregated score was created by summing up all the answers.

Sociodemographic variables. All participants also responded to sociodemographic questions about their age, gravidity, educational qualifications (illiterate, primary, secondary, university degree) of themselves and their husbands, job status (employed, unemployed, student), history of addiction for themselves and their husbands, number of children, and having undergone RT (yes/no). 

The draft questionnaire was piloted among 40 health professionals and pregnant women. Based on the participants’ viewpoints, ambiguous items were changed. Appropriateness and reliability [25] were determined for the questionnaire. The reliability test for internal consistency for all subscales was good, with Cronbach’s α coefficients ranging from 0.76 to 0.93.

### 2.4. Ethics

Ethical approval was obtained from the ethics committee of Tabriz University of Medical Sciences (IR. TBZMED.REC.1396.689). All participants were informed about the purpose and method of the study. Informed written consent was obtained from all participants.

### 2.5. Statistical Analysis

Descriptive statistics were used for the demographic characteristics and the knowledge, attitudes, and behavior scales related to HIV prevention and HIV RT. To investigate the relationship between participants’ sociodemographic characteristics and knowledge, attitudes, and behaviors, a multivariate general linear model (GLM) was applied. GLM using the “main effect” was performed to find predictors (independent factors: age, husbands’ age, education, husbands’ education, gravidity, and job associated with knowledge, attitudes, and behaviors related to HIV prevention and RT [dependent variable]) and to yield the odds ratio (OR) and confidence interval (CI 95%). 

## 3. Results

Characteristics of the Participants

A total of 200 pregnant women referred to marginal health centers to obtain pregnancy healthcare participated. Demographic characteristics of the participants are given in Table 1. The mean age (SD) was 27.83 (SD = 5.8), and about half of the participants had a high school education level. With regard to gravidity, the average family size was 3.38. We divided gravidity into two or fewer than two and three or more than three pregnancies. The average marriage duration was 6.1 years. Four people had had a positive HIV rapid test, and one of them had also had a positive diagnostic test. 

Table 2 shows the knowledge and attitudes of pregnant women regarding HIV prevention and HIV RT. Regarding transmission mode, 87.5% and 59.5% of pregnant women declared that HIV could not be transmitted from hand shaking, social kissing, or hugging, and food or water, respectively. Nearly all of the participants (97%) knew that HIV can be transmitted through sexual intercourse. Although a majority of the pregnant women knew that mother-to-child HIV transmission is possible during pregnancy, fewer than half (48.2%) of them knew that HIV can be transmitted from mother to child through breastfeeding. Younger pregnant women (under 26 years old) had a significantly higher rate of HIV transmission through social relationships (χ^2^(1) = 3.61, *p* < 0.05). Additionally, about half (54.5%) of the participants falsely believed that all babies born to HIV-positive mothers are infected with HIV. Only 22.5% of pregnant women knew about the recommendation of a Cesarean section for HIV-positive mothers. About 75.5% and 69% of participants knew about how (i.e., free tests within the health centers) and when (i.e., the first time they were referred to the centers) HIV RT is generally conducted, respectively, but only 32% knew about how many times HIV RT can be carried out. 

The mean (SD) and median (interquartile range (IQR)) values of attitudes of pregnant women regarding HIV prevention and HIV RT were 4.5 (0.4) and 51 (7.75), respectively. The attitude toward the HIV prevention dimension was normally distributed; therefore, the mean (SD) and *t*-test were used for reporting results; because the attitudes toward RT were also not normally distributed, we used median and IQR and Mann–Whitney tests for reporting results.

Table 3 presents the behaviors related to HIV prevention and HIV RT among pregnant women. The behaviors addressed in this study and the rate of respondents were the following: undergoing HIV RT at any time (94.5%), undergoing HIV RT (97%) or HIV diagnostic testing during pregnancy (18%), informing their family about HIV RT results (86%), participating in HIV counseling classes (25.5%) or HIV RT counseling classes (11.5%), high-risk behavior (smoking [10%] and opium or drug use [2.5%]), and making a decision about Cesarean section after an HIV-positive test (41.5%). 

Potential predictors of knowledge, attitudes, and behaviors regarding HIV prevention and HIV RT are shown in Table 4. Compared to women who had academic education, illiterate pregnant women were 12.6 times more likely to have poor knowledge of HIV. Additionally, women whose husbands were illiterate were 6.4 times more likely to have poor knowledge, 9.2 times more likely to have poor attitudes, and 2.3 times more likely not to exhibit HIV prevention behaviors. The other variables did not show significant associations in the GLM analysis. 

## 4. Discussion

Improving knowledge, attitudes, and behaviors regarding HIV among reproductive populations is essential in comprehensive HIV prevention programs, especially among pregnant women who live in marginal areas where there is a higher risk of HIV infection. The findings of the present study indicate that pregnant women have adequate knowledge of HIV prevention in general, but this was not the case for knowledge about mother-to-child HIV transmission. The results indicate that pregnant women do not have the essential knowledge to prevent HIV transmission to children. In this study, 97% of the participants knew that sexual intercourse is a major mode of transmission, but in terms of the vital aspects of HIV transmission among pregnant women, a lack of information was observed, such as the fact that a Cesarean section is recommended for HIV-positive pregnant women. This result corresponds with findings of research conducted in China, Ghana, and Ethiopia, in which only 57%, 25%, and 34.9% of women, respectively, knew about mother-to-child HIV transmission [23,26,27,28,29,30]. The poor knowledge of mother-to-child transmission in this study can be attributed to the lack of proper counseling and poor health education in the health delivery centers, especially regarding the three vital modes of HIV transmission from mother to child during pregnancy, childbirth, and breastfeeding. As our results show, only 11.5% of women in this study had participated in counseling classes for RT, and 25.5% had participated in HIV education. 

In a recent report, the Iranian Ministry of Health warned that there has been about a threefold increase in sexual transmission during the past decades [31]. Based on this report, Iran has committed to designing a plan with fast-track targets by collaborating with UNAIDS to reach a world with zero new HIV infections and deaths by the year 2030 [11]. In order to achieve this goal, and according to the international program, people with HIV should know their status and receive related treatment [32]. Therefore, pregnant women in marginal regions with high-risk conditions for infection and low sociodemographic characteristics are one of the most important risk groups for targeted HIV prevention. 

Although the Iranian Ministry of Health has taken progressive steps toward HIV prevention, there are no education campaigns, and a lack of proper education even in health centers has made HIV prevention programs complicated [11]. The WHO’s Policy Statement regarding HIV testing strongly recommends counseling before and after testing [33].

About 70% of pregnant women in this study were young (15–25 years), and most of the women had completed high school. In terms of sociodemographic factors related to the knowledge, attitudes, and behaviors of pregnant women regarding HIV prevention and HIV RT, there was a significant negative association between a lack of education and knowledge, attitudes, and behaviors. The results of this study suggest that socio-demographic factors such as education are the main determinants of the knowledge related to HIV prevention and HIV RT among pregnant women living in marginal areas. The importance of education for improving health-related outcomes has been reported in other studies [34,35]. In fact, social factors such as education are fundamental determinants of HIV prevention because they determine access to resources such as income, safe environments, and healthy lifestyles. Education promotes individual’s knowledge, skills, perceptions, and a broad range of other abilities related to disease prevention [36,37]. 

### Strength and Limitation 

An important strength of the study was its setting and data collection from marginal areas of Kermanshah, a city with the highest prevalence of HIV in Iran. However, because the results of this study were derived from a high-risk marginal area in Iran, they may not be generalizable to populations with a different socioeconomic status. 

## 5. Conclusions

The findings of this study highlight the need for education and counseling classes and campaigns to improve knowledge, attitudes, and behaviors related to HIV prevention, especially during pregnancy among women in marginal regions. Proper counseling for women with the intention of becoming pregnant and providing essential knowledge in terms of the vital aspects of HIV transmission during pregnancy, childbirth, and breastfeeding will be critical to substantially decrease HIV transmission. 

## Figures and Tables

**Table 1 healthcare-09-00793-t001:** Demographic characteristics of pregnant women in the marginal area of Kermanshah, Iran (*n* = 200).

Variables	*N* (%)
Age Mean (SD)	27.83 (5.8)
Husband’s age Mean (SD)	32.08 (6.6)
Education level	
Illiterate	8 (4.0)
High school	105 (53.0)
Diploma	69 (34.5)
Academic	17 (8.5)
Husband’s education	
Illiterate	9 (4.5)
High school	104 (52.0)
Diploma	73 (36.5)
Academic	14 (7.0)
Gravidity	
2≤	141 (70.5)
3≥	59 (29.5)
History of high-risk behavior (yes)	2 (1.0)
Husband with a history of high-risk behavior (yes)	9 (4.5)
Family size Mean (SD)	3.38 (1.4)
Number of children Mean (SD)	0.71 (0.8)
Marriage duration (years) Mean (SD)	6.10 (4.9)
Positive HIV rapid test *	4 (2.0)
Positive HIV diagnosis **	1 (0.5)
Husband with HIV^+^ ***	1 (0.5)

Note: * Pregnant women who were screened by rapid testing and were positive for HIV. ** Infected pregnant women who were positive in rapid testing and confirmed by an enzyme-linked immunosorbent assay (ELISA) diagnosis test. *** Pregnant women whose husbands were infected with HIV.

**Table 2 healthcare-09-00793-t002:** Percentage of correct answers of pregnant women on the knowledge scale and mean scores on the attitude scale about HIV prevention and HIV RT.

	Age			
	Total (*n* = 200)	15–25 (*n* = 137)	26 ≥ (*n* = 63)	*p*-Value
Knowledge (yes or no)				
Can HIV be transmitted from:				
Hand shaking, social kissing, or hugging	175 (87.5)	124 (90.5)	51 (81.0)	0.049
Food or water	119 (59.5)	81 (59.1)	38 (60.3)	0.873
Sexual intercourse	194 (97.0)	132 (96.4)	62 (98.4)	0.427
Mother to her child during pregnancy	165 (82.5)	110 (80.3)	55 (87.3)	0.226
Mother to her child through breastfeeding	96 (48.2)	66 (48.2)	30 (47.6)	0.942
Can pregnant women be infected with HIV?	152 (76.0)	104 (75.9)	48 (76.2)	0.999
Are all babies born to HIV-positive mothers infected with HIV?	109 (54.5)	75 (54.7)	34 (54.0)	0.918
Does using a condom reduce the risk of HIV?	134 (67.0)	91 (66.4)	43 (68.3)	0.798
Is a Cesarean section recommended for HIV-positive women?	45 (22.5)	27 (19.7)	18 (28.6)	0.163
How can HIV RT be conducted?	151 (75.5)	105 (76.6)	46 (73.0)	0.580
How many times can HIV RT be carried out?	64 (32.0)	40 (29.2)	24 (38.1)	0.254
When is HIV RT usually done during pregnancy?	138 (69.0)	94 (68.6)	44 (69.8)	0.871
Is breastfeeding recommended for HIV-positive mothers?	99 (49.5)	70 (51.1)	29 (46.0)	0.506
Total knowledge (mean; SD)	8.2 (2.1)	8.1 (2.2)	8.3 (2.1)	0.724
Attitudes				
Attitudes toward HIV prevention (Mean, SD)	4.5 (0.4)	3.9 (0.4)	4.0 (0.4)	0.048 ^b^
Attitudes toward HIV RT (Median; IQR ^a^)	51 (7.75)	50 (7)	53 (7)	0.051 ^c^

Note: ^a^: interquartile range; ^b^: derived from independent *t*-test, ^c^: derived from Mann–Whitney test.

**Table 3 healthcare-09-00793-t003:** Behaviors related to HIV prevention and RT among pregnant women in the marginal area of Kermanshah, Iran.

	Age			
Behavior (Yes)	Total (*n* = 200)	15–25 (*n* = 137)	26 ≥ (*n* = 63)	*p*-Value
Were you ready to undergo HIV RT at any time?	189 (94.5)	60 (95.2)	189 (94.5)	0.756
Have you undergone HIV RT during pregnancy?	195 (97.5)	134 (97.8)	61 (96.8)	0.679
Did you inform your family of HIV RT results?	172 (86.0)	115 (83.9)	57 (90.5)	0.216
Husband	167 (83.5)	113 (82.2)	54 (85.7)	
Mother	4 (2.0)	2 (3.2)	2 (1.5)	
Sister or brother	1 (0.5)	0 (0.0)	1 (1.6)	
Have you ever participated in HIV RT counseling classes?	23 (11.5)	6 (9.5)	17 (12.4)	0.552
Have you ever had high-risk behavior?	15 (7.5)	8 (5.8)	7 (11.1)	0.552
Smoking	10 (5.0)	5 (3.6)	5 (7.9)	
Opium or drug use	5 (2.5)	3 (3.0)	2 (3.1)	
Have you ever undergone a diagnostic HIV test?	36 (18.0)	25 (18.2)	11 (17.5)	0.893
Will you prefer a Cesarean section for your baby if you are HIV-positive?	83 (41.5)	37 (58.7)	83 (41.5)	0.001
Will you take the medicine if you are HIV-positive?	155 (77.5)	103 (75.2)	52 (82.5)	0.247
Have you ever participated in HIV education and counseling sessions?	51 (25.5)	33 (24.1)	18 (28.6)	0.499

Note: *p*-value derived from chi-squared tests.

**Table 4 healthcare-09-00793-t004:** Results of multivariate GLM analysis investigating potential predictors of knowledge, attitudes, and behaviors regarding HIV prevention and HIV RT.

Variables	Knowledge		Attitudes		Behaviors	
	OR (95% CI)	P	OR (95% CI)	P	OR (95% CI)	P
Education level						
Illiterate	−12.6 (−18.0; −7.3)	<0.001	2.6 (−5.4; 10.6)	0.524	0.33 (−1.8; 2.5)	0.758
High school	−9.1 (−12.2; −5.9)	<0.001	−1.14 (−5.8; 3.6)	0.637	−1.36 (−2.6; −0.09)	0.036
Diploma	−5.6 (−8.8; −2.4)	0.001	−0.15 (−5.0; 4.6)	0.949	−0.83 (−2.2; 0.47)	0.210
Academic	Ref		Ref		Ref	
Husband’s education						
Illiterate	−6.4 (−11.8; −1.1)	0.022	−9.2 (−17.4; −1.1)	0.027	−2.3 (−4.5; −1.1)	0.041
High school	−2.3 (−5.8; 1.4)	0.210	−4.2 (−5.4; 10.6)	0.524	−1.1 (−2.6; 0.29)	0.117
Diploma	−2.0 (−5.6; 1.5)	0.255	−0.157 (−5.0; 4.6)	0.949	−0.76 (−2.2; 0.68)	0.298
Academic	Ref		Ref		Ref	
Age	−0.014 (−0.24; 0.21)	0.902	0.118 (−0.228; 0.463)	0.503	0.012 (−0.08; 0.104)	0.805
Gravidity						
2≤	−1.7 (−4.0; 0.48)	0.122	−1.4 (−4.8; 1.8)	0.386	−0.46 (−1.3; 0.45)	0.321
3≥	Ref		Ref	Ref		

## Data Availability

The data collection tools and datasets generated and/or analyzed during the current study are available from the corresponding author on reasonable request.

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
