# Peer review of "A Study on the Knowledge, Attitudes, and Behaviors of Pregnant Women Regarding HIV and Routine Rapid Testing: An Assessment in a High-Risk Marginal Area"

_healthcare, 2021, doi:10.3390/healthcare9070793_

Round 1
Reviewer 1 Report
Thank you for sending the review request on the study title "A study on the knowledge, attitudes, and behaviors of pregnant women regarding HIV and routine rapid testing: an assessment in a high-risk marginal area". Here are some of my suggestions that can improve the scientific quality of this draft.
Abstract:
Write the full name of AIDS at the start of the abstract and then write in abbreviation
Line #14-15. Write this sentence like this. Participants and pregnant women both are written in the sentence that needs to be revised. Pregnant women who were referred to the local health centers and who were willing to participate in the study were interviewed.
The method section of the abstract explains the research design. Moreover, what is a paper-pencil questionnaire? Furthermore, don't write (mean age = 27.8, SD = 5.8) in the method section of abstract. You may write this in the result section.
L#25 (range = 1 - 5) write what is meaning of 1 and 5 hereto quickly understand the context of the findings.
Introduction
The introduction is well written I would only suggest identifying the gaps of previous studies and then make the case to conduct this study. Moreover, the authors can write the justification for designing this study in the introduction section.
Method section:
Study design:
The study design is missing in this section. Please clearly mention which study design was employed in this research and why this specific design is the best suited for this study.
L#125. Based on the participants’ viewpoints, ambiguous items were adapted. Adapted, changed, deleted, or rephrased??
Sample Size:
It also adds value if the authors mention the scientific justification of calculating sample size.
Results:
Table 01
Positive HIV rapid test 4 (2.0)
Positive HIV diagnosis 1 (0.5)
Husband with HIV 1 (0.5) +
What is the meaning of these statements? How the readers will be able to understand the context of these findings? Use the appropriate footer at the end of the table to explain some key terms.
The rule of thumb is a measure of central tendency is: mean and standard deviation is not the appropriate analysis technique for 5 points Likert scale. Please check and use appropriate references for this analysis.
Discussion:
The discussion section needs to be improved by contextualizing the study findings with relevant literature. For example the last para L#228-231 ONLY presenting the findings of the study. However, it needs to be discussed why "negative association between lack of education and knowledge, attitudes, and behaviors. "
Conclusions:
The conclusion is vague that does not present the policy recommendations. This section needs to improve to suggest some policy implications for policymakers.
Moreover, the authors should write the strengthens and limitations of this study.
Author Response
Thank you for the opportunity to address the revisions recommended by the reviewers for the manuscript "A study on the knowledge, attitudes, and behaviors of pregnant women regarding HIV and routine rapid testing: an assessment in a high-risk marginal area". We would like to thank the reviewers for the time and the helpful feedbacks which enabled us to improve our manuscript.
In line with the reviewers’ suggestions, we have edited the manuscript for clarity. For detailed information on how we have addressed each of the comments, we would like to refer to the letter below. In order to indicate the revisions made, we used colored text to highlight the changes in the revised version of our manuscript.
Thank you again for your valuable and well-appreciated comments.
Reviewer 1
Comments and Suggestions for Authors
Thank you for sending the review request on the study title "A study on the knowledge, attitudes, and behaviors of pregnant women regarding HIV and routine rapid testing: an assessment in a high-risk marginal area". Here are some of my suggestions that can improve the scientific quality of this draft.
Abstract:
Write the full name of AIDS at the start of the abstract and then write in abbreviation.
**answer to reviewer: Thank you. We have changed this.
Line #14-15. Write this sentence like this. Participants and pregnant women both are written in the sentence that needs to be revised. Pregnant women who were referred to the local health centers and who were willing to participate in the study were interviewed.
**answer to reviewer: Thank you. This was revised in the following way:
“Pregnant women who were referred to the local health centers and who were willing to participate in the study were interviewed by using a standard questionnaire.”
The method section of the abstract explains the research design. Moreover, what is a paper-pencil questionnaire? Furthermore, don't write (mean age = 27.8, SD = 5.8) in the method section of abstract. You may write this in the result section.
**answer to reviewer: Thank you for your helpful suggestion. This has been changed.
L#25 (range = 1 - 5) write what is meaning of 1 and 5 hereto quickly understand the context of the findings.
**answer to reviewer: This was deleted and described in the result section.
Introduction
The introduction is well written I would only suggest identifying the gaps of previous studies and then make the case to conduct this study. Moreover, the authors can write the justification for designing this study in the introduction section.
**answer to reviewer: Thank you, the following sentence was added according to your suggestion.
“There are limited studies regarding HIV-related knowledge, RT behaviors and beliefs among high risk population groups. A study from US among internal medicine found that less than one third of physicians always or usually performing routine testing [16]. “
Method section:
Study design:
The study design is missing in this section. Please clearly mention which study design was employed in this research and why this specific design is the best suited for this study.
**answer to reviewer: The study was a cross-sectional study. In the revised version of the paper, we mention this in the method section: “A cross-sectional design was applied, given that we wanted to assess the knowledge and attitudes as diseases traits.”
L#125. Based on the participants’ viewpoints, ambiguous items were adapted. Adapted, changed, deleted, or rephrased?
**answer to reviewer: Thank you, done.
Sample Size:
It also adds value if the authors mention the scientific justification of calculating sample size.
**answer to reviewer: Thank you for your valuable recommendation. The following part was added to the method section (2.2. Study design and sample size):
“The sample size for the study was calculated using the following formula (n=Z2(1-α/2) P (1-P)/d2) where z(1-α/2) for 95% confidence interval was 3.84, P= 0.6 [18], and d=0.72. As such, based on the estimated sample size, at least 185 pregnant woman were required for this study to have a power of 80% at 5% significant level. However, in practice 200 pregnant women enrolled to the study.”
Results:
Table 01
Positive HIV rapid test 4 (2.0)
Positive HIV diagnosis 1 (0.5)
Husband with HIV 1 (0.5) +
What is the meaning of these statements? How the readers will be able to understand the context of these findings? Use the appropriate footer at the end of the table to explain some key terms.
**answer to reviewer:
We inserted footnotes
Note:
*Pregnant women who were screened by rapid testing and were positive for HIV.
** Infected pregnant women who were positive in rapid testing and confirmed by enzyme-linked immunosorbent assay (ELISA) diagnosis test.
***pregnant women who their husbands were infected with HIV
Table 2:
Note: a: interquartile range; b: derived from independent t-test, c: derived from Man-Withney test.
The rule of thumb is a measure of central tendency is: mean and standard deviation is not the appropriate analysis technique for 5 points Likert scale. Please check and use appropriate references for this analysis.
**answer to reviewer: Many thanks for your comment. This part was revised and re-analyzed according to your recommendation. As the attitude toward HIV prevention dimension was normally distributed, mean (SD) and T-test were used for reporting results, and also as the attitude toward RT was not normally distributes, we used median and IQR and Man-Withney tests for reporting results.
Discussion:
The discussion section needs to be improved by contextualizing the study findings with relevant literature. For example the last para L#228-231 ONLY presenting the findings of the study. However, it needs to be discussed why "negative association between lack of education and knowledge, attitudes, and behaviors.
**answer to reviewer: Thank you very much. The following part was added to the last paragraph in the discussion section.
“The results of this study suggest that socio-demographic factors such as education are the main determinants of the knowledge related to HIV prevention and HIV-RT among pregnant women living in marginal areas. The importance of education for improving health related outcomes had been reported in other studies [31-32]. In fact, social factors such as education are fundamental determinant of HIV prevention because they determine access to resources like income, safe environment, and healthy lifestyle. Education promotes individual’s knowledge, skills, perceptions, and a broad range of other abilities related to diseases prevention [33-34]. “
Conclusions:
The conclusion is vague that does not present the policy recommendations. This section needs to improve to suggest some policy implications for policymakers.
**answer to reviewer: The following part was added:
“Proper counselling for women with the intention to become pregnant and providing essential knowledge in terms of the vital aspects of HIV transmission during pregnancy, childbirth, and breastfeeding will be critical to substantially decrease HIV transmission. “
Moreover, the authors should write the strengthens and limitations of this study.
**answer to reviewer: We have added the following sentences: “An important strength of the study was its setting and data collection from marginal areas of Kermanshah as a city with highest prevalence of HIV in Iran. The results of this study were derived from high risk marginal area from Iran and may not be generalizable to population with different socioeconomic status. “

Reviewer 2 Report
Thank you for your submission. My concerns are as follows-
- Your sample is a convenience sample, but still, there needs to be some type of power analysis or sample size estimation procedure.
- The measures are not valid and reliable as they have not been tested or pilot trialed.
- The analyses is mostly descriptive with no inferential statistics used.
- The topic has been extensively studied, no new knowledge is being added.
Author Response
Thank you for the opportunity to address the revisions recommended by the reviewers for the manuscript "A study on the knowledge, attitudes, and behaviors of pregnant women regarding HIV and routine rapid testing: an assessment in a high-risk marginal area". We would like to thank the reviewers for the time and the helpful feedbacks which enabled us to improve our manuscript.
In line with the reviewers’ suggestions, we have edited the manuscript for clarity. For detailed information on how we have addressed each of the comments, we would like to refer to the letter below. In order to indicate the revisions made, we used colored text to highlight the changes in the revised version of our manuscript.
Thank you again for your valuable and well-appreciated comments.
Comments and Suggestions for Authors:
Thank you for your submission. My concerns are as follows-
- Your sample is a convenience sample, but still, there needs to be some type of power analysis or sample size estimation procedure.
**answer to the reviewer: Thank you for your valuable recommendation. The following part was added to the method section (2.2. Study design and sample size):
The sample size for the study was calculated using the following formula (n=Z2(1-α/2) P (1-P)/d2) where z(1-α/2) for 95% confidence interval was 3.84, P= 0.6 [18], and d=0.72. As such the sample size was estimated at least 185 pregnant woman requiring for the study to have a power of 80% at 5% significant level. However, in practice 200 pregnant women enrolled to the study.
- The measures are not valid and reliable as they have not been tested or pilot trialed.
**answer to reviewer: Thank you very much for your comment. This part was added: “The dimensions of the questionnaire were qualitatively assessed based on a conceptual framework by five health professional’s panel. The health professional panel determined if the content of the questionnaire was relevant to the conceptual framework. The attitude dimension was evaluated based on the validity and reliability indexes before data collection, and found to be valid and reliable measure. The Cronbach’s alpha coefficients for the separate dimensions ranged from 0.65 to 0.89. Additionally, the reliability of the dimensions assessed by Intraclass Correlation Coefficient (ICC) and showed satisfactory results (ICC ranged from 0.75 to 0.82) among (n = 10) pregnant women. The questionnaire was pilot tested with a sample of 20 respondents.”
- The analyses is mostly descriptive with no inferential statistics used.
**answer to reviewer: Table 4 shows the results of multivariate GLM analysis investigating potential predictors of knowledge, attitudes, and behaviors regarding HIV prevention and HIV RT.
- The topic has been extensively studied, no new knowledge is being added.
**answer to reviewer: We have added in the discussion section the strength of this study: “An important strength of the study was its setting and data collection from marginal areas of Kermanshah as a city with highest prevalence of HIV in Iran. The results of this study were derived from high risk marginal area from Iran and may not be generalizable to population with different socioeconomic status.“
Reviewer 3 Report
Thank you for sending your paper entitled “A study on the knowledge, attitudes, and behaviors of pregnant women regarding HIV and routine rapid testing: an assessment in a high-risk marginal área” to Healthcare. After carefully review this interesting paper, the following comments are listed for your reference:
- Abstract: To increase potential citations, authors should check keywords against those recommended in the MeSH Browser of Medical Subject Headings https://meshb.nlm.nih.gov/. For example: “rapid testing is not MeSH. I recommend that you change this keyword.
- Abstract (lines 12, 16, 20 and 26): According to the norms of the journal in the abstract it is not necessary to put the sections, for that reason they eliminate “Background”, “Methods”,”Results” and “Conclusion”.
- Introduction (line 41): review this data and correct it “At the end of 2106”.
- Introduction: In the introduction section, they should add what the scientific evidence says about knowledge, attitudes and behavior of pregnant women regarding HIV and routine rapid testing. What is known about the subject? Are there recent studies on this topic?
- Result: Under each table you should indicate the statistical analysis by which the p-value is obtained.
- Discussion: Within the discussion, a subsection "limitations" of the study should be added. Please add it
- Discussion: In general, I see the discussion very brief. There is a lack of bibliographic support for your results with recent literature (impact articles). Please reinforce this section.
- References: The bibliography used is too old. Please update it and add recent articles related to your results. 18/30 are over 10 years old and 28/30 are over 5 years old.
- References: Review the journal's bibliography citation guidelines. Your bibliography is not well written
Author Response
Thank you for the opportunity to address the revisions recommended by the reviewers for the manuscript "A study on the knowledge, attitudes, and behaviors of pregnant women regarding HIV and routine rapid testing: an assessment in a high-risk marginal area". We would like to thank the reviewers for the time and the helpful feedbacks which enabled us to improve our manuscript.
In line with the reviewers’ suggestions, we have edited the manuscript for clarity. For detailed information on how we have addressed each of the comments, we would like to refer to the letter below. In order to indicate the revisions made, we used colored text to highlight the changes in the revised version of our manuscript.
Thank you again for your valuable and well-appreciated comments.
Thank you for sending your paper entitled “A study on the knowledge, attitudes, and behaviors of pregnant women regarding HIV and routine rapid testing: an assessment in a high-risk marginal área” to Healthcare. After carefully review this interesting paper, the following comments are listed for your reference:
- Abstract:To increase potential citations, authors should check keywords against those recommended in the MeSH Browser of Medical Subject Headings https://meshb.nlm.nih.gov/. For example: “rapid testing is not MeSH. I recommend that you change this keyword.
**answer to reviewer: Thank you. The keywords were revised based on the introduced Mesh site.
- Abstract (lines 12, 16, 20 and 26): According to the norms of the journal in the abstract it is not necessary to put the sections, for that reason they eliminate “Background”, “Methods”,”Results” and “Conclusion”.
**answer to reviewer: These sections were deleted.
- Introduction (line 41): review this data and correct it “At the end of 2106”.
**answer to reviewer: Thank you, it was revised.
- Introduction: In the introduction section, they should add what the scientific evidence says about knowledge, attitudes and behavior of pregnant women regarding HIV and routine rapid testing. What is known about the subject? Are there recent studies on this topic?
**answer to reviewer: Thank you. We have changed this in the following way:”
There are limited studies regarding HIV-related knowledge, RT behaviors and beliefs among high risk population groups. A study from US among internal medicine found that less than one third of physicians always or usually performing routine testing [16].”
- Result:Under each table you should indicate the statistical analysis by which the p-value is obtained.
**answer to reviewer: Thank you, done. Have inserted footnotes.
Note: *pregnant women who screened by rapid testing and were positive for HIV.
** Infected pregnant women who were positive in rapid testing and confirmed by enzyme-linked immunosorbent assay (ELISA) diagnosis test.
***pregnant women who their husbands were infected with HIV
Table 2: Note: a: interquartile range; b: derived from independent t-test, c: derived from Man-Wittney test.
- Discussion: Within the discussion, a subsection "limitations" of the study should be added. Please add it.
**answer to reviewer: Thank you for your valuable comment: as you mentioned the following sentences were added.
“Strength and limitation
An important strength of the study was its setting and data collection from marginal areas of Kermanshah as a city with highest prevalence of HIV in Iran. The results of this study were derived from high risk marginal area from Iran and may not be generalizable to population with different socioeconomic status. “
- Discussion: In general, I see the discussion very brief. There is a lack of bibliographic support for your results with recent literature (impact articles). Please reinforce this section.
Thank you. The following part was added:
The results of this study suggest that socio-demographic factors such as education are the main determinants of the knowledge related to HIV prevention and HIV-RT among pregnant women living in marginal areas. The importance of education for improving health related outcomes had been reported in other studies [31-32]. Infact, social factors such as education are fundamental determinant of HIV prevention because they determine access to resources like income, safe environment, and healthy lifestyle. Education promote individual’s knowledge, skills, perceptions, and a broad range of other abilities related to diseases prevention [33-34].
- References:The bibliography used is too old. Please update it and add recent articles related to your results. 18/30 are over 10 years old and 28/30 are over 5 years old.
**answer to the reviewer: Done.
- References:Review the journal's bibliography citation guidelines. Your bibliography is not well written
**answer to reviewer: Done.
Round 2
Reviewer 1 Report
Thank you so much for sharing the revised draft for review. I have carefully reviewed the changes made by the authors. It seems fine to me and this article may be published in its current form.
Reviewer 2 Report
Thank you for the revisions.
Reviewer 3 Report
Accept in present form.